# Nanostructured Scaffold, Combined with Human Dental Pulp Stem Cell Secretome, Induces Vascularization in Medicinal Leech Model

**DOI:** 10.3390/mi16101150

**Published:** 2025-10-10

**Authors:** Gaia Marcolli, Nicolò Baranzini, Ludovica Barone, Federica Rossi, Laura Pulze, Christina Pagiatakis, Roberto Papait, Annalisa Grimaldi, Rosalba Gornati

**Affiliations:** 1Department of Biotechnology and Life Sciences, University of Insubria, 21100 Varese, Italy; gmarcolli@uninsubria.it (G.M.); nicolo.baranzini@uninsubria.it (N.B.);; 2Department of Cardiovascular Medicine, IRCCS Humanitas Research Hospital, 20089 Rozzano, Italy

**Keywords:** nanostructured scaffold, human Dental Pulp Stem Cells, secretome, tissue regeneration, cell-free device, medicinal leech

## Abstract

As life expectancy continues to increase, age-related disorders are becoming more prevalent. Among these, vascular complications resulting from chronic inflammation are particularly concerning, as they impair angiogenesis and hinder tissue repair, both processes that heavily rely on a well-structured extracellular matrix (ECM). In this context, MicroMatrix^®^ UBM Particulate, a skin substitute composed of collagen, laminin, and proteoglycans, appears to offer properties conducive to tissue regeneration. The aim of this study was to evaluate the regenerative potential of MicroMatrix^®^ combined with the Secretome of human Dental Pulp Stem Cells (hDPSC-S), using the medicinal leech *Hirudo verbana*, a well-established model for studying wound healing, angiogenesis, and tissue regeneration. Adult leeches were injected with MicroMatrix^®^ either suspended in FBS-free medium (CTRL) or supplemented with hDPSC-S. 1-week post-treatment, the animals were sacrificed and subjected to morphological and immunohistochemical analyses. Our findings revealed that MicroMatrix^®^ successfully integrated into the leech body wall. Notably, when supplemented with hDPSC-S, there was a marked increase in cell infiltration, including telocytes and Hematopoietic Precursor Stem Cells, along with a significantly higher vessel density compared to CTRL. These results support the effectiveness of the cell-free device composed of MicroMatrix^®^ and hDPSC-S, highlighting its potential as a promising strategy for regenerative therapies aimed at treating complex wounds with poor vascularization.

## 1. Introduction

As life expectancy increases, age-related disorders are becoming more prevalent. Among them, vascular complications caused by chronic inflammation are associated with impaired angiogenesis and tissue repair [1,2]. Indeed, diabetic foot ulcers occur in approximately 15% of patients with diabetes, with about 6% requiring hospitalization for infections [3,4,5].

Thus, research on revascularization and wound-healing treatments remains both a challenge and an unmet clinical need [6,7]. It is well established that, following tissue injury, angiogenesis is essential to ensure adequate oxygen and nutrient delivery for tissue restoration [8,9,10]. This process relies on the presence of a proper extracellular matrix (ECM), in terms of composition, stiffness, and growth factor content, which guides Endothelial Cell (EC) activation and proliferation, necessary for vessel formation and stabilization [11,12]. Therefore, one of the major aims of regenerative medicine is the development of an ECM substitute [13,14,15].

In this context, a nanostructured three-dimensional scaffold serves as a framework that preserve the three-dimensional organization of the ECM, providing a defined microenvironment that directs vascular cell proliferation, migration, and adhesion. In the last decade, plenty of smart nanostructured biomaterials (polymeric, ceramic, composite, hydrogels) have been engineered to respond to environmental stimuli, control drug or growth factor release, and precisely match mechanical or structural tissue requirements [16,17]. Compared to these, MicroMatrix^®^ UBM Particulate (here referred as MicroMatrix^®^) offers the advantage of biological authenticity, being composed of an extracellular matrix (ECM) derived from porcine urinary bladder. It has been shown to integrate well and be well tolerated in vivo. Applied either as a powder or a paste, it maintains intimate contact with all areas of the wound bed, offering an optimal solution for managing irregular wounds. Furthermore, it has been shown to support a shift from an inflammatory wound environment to one that facilitates rapid revascularization and promotes wound closure. In addition, the association of the scaffold with soluble factors, such as those present in the Mesenchymal Stem Cell Secretome (MSC-S), has proven to be more effective in supporting in vivo angiogenesis, a minimal but necessary condition for promoting the tissue regeneration process [18,19,20,21].

Among the various MSC populations, human Dental Pulp Stem Cells (hDPSCs) have garnered significant interest due to their high proliferative capacity and the maintenance of stemness properties at least until the 30th passage [21]. Additionally, they can be easily obtained from the extraction of impacted wisdom teeth, which are typically discarded as surgical waste [20,22,23]. These characteristics, along with the ability of hDPSC Secretome (hDPSC-S) to enhance ECM remodelling, accelerate angiogenesis, and support muscle regeneration, key processes in tissue repair, make hDPSCs an attractive source for regenerative therapies.

Building on previous findings, this study investigates the effects of MicroMatrix^®^, supplemented with hDPSC-S, in the medicinal leech *Hirudo verbana*, a well-established model for wound healing, angiogenesis, and tissue repair. Indeed, the leech offers distinct advantages as a model organism for studying tissue repair and immune activation, due to its small size, well-defined anatomy, and well-characterized cell populations. Unlike rodent models, it enables real-time observation of regenerative and immune events in vivo, with reduced biological complexity, minimal maintenance costs, and no interference from adaptive immune mechanisms [24]. During the regenerative phases, following tissue injury, the reorganization of connective tissue is orchestrated by soluble mediators such as Endothelial Growth Factor (EGF), basic Fibroblast Growth Factor (bFGF), and cathepsin B, which are secreted by fibroblasts and immune cells. These factors play a pivotal role in stimulating fibroblast proliferation and promoting collagen synthesis. Notably, the immune and regenerative mechanisms observed in the leech show remarkable parallels with those of vertebrates, relying on conserved molecular pathways and comparable cellular players [24]. Furthermore, previous studies show that scaffold enriched with Vascular Endothelial Growth Factor (VEGF) foster the recruitment of CD31^+^ endothelial cells and CD34^+^ Hematopoietic Precursor Stem Cells (HPSCs), involved in angiogenesis and muscle regeneration [24,25,26].

Within this biological framework, this study contributes to establishing the potential of a cell-free therapeutic strategy consisting of the nanostructured scaffold MicroMatrix^®^, combined with hDPSC-S, in enhancing ECM remodelling, which in turn accelerates angiogenesis and muscle regeneration, ultimately resulting in an optimal device for clinical applications.

## 2. Materials and Methods

### 2.1. Ethics and Consent

Human Dental Pulp Stem Cells (hDPSCs) were isolated from dental pulp of a healthy subject (female, 11 years old), undergoing third molar extraction. The subject provided the informed consent, to be included in the study, had no history of metabolic disorders, and was not taking medications at the time of the surgical procedure. The study was approved by the institutional review board ethics committees. Subjects were recruited within a clinical protocol by “Ospedale di Circolo Fondazione Macchi” and approved by the institutional Ethical Committee (Protocol number: 0034066, approval date: 3 October 2013) according to the Helsinki Declaration of 1975, as revised in 2013.

In vivo studies have been conducted using medicinal leeches of the species *H. verbana*, which are not protected under Directive 2010/63/EU governing the use of animals in scientific research, nor are they listed in Legislative Decree no. 26, of 4 March 2014, “Implementing Directive 2010/63/EU on the protection of animals used for scientific purposes”, published in the Italian Official Journal on 14 March 2014. However, their use has been authorized by the Animal Welfare Body (OPBA) at the University of Insubria.

### 2.2. Materials and Reagents

MicroMatrix^®^ was donated by LifeSciences Corporation (Plainsboro, NJ, USA). Chemicals were purchased from Sigma-Aldrich (Milano, Italy), Promega (Milano, Italy), Zymo Research (Milan, Italy), BioRad (Milano, Italy), Invitrogen-Life Technologies (Milano, Italy), Bio Optica (Milan, Italy) unless otherwise indicated. Amicon Ultra 15 mL Centrifugal filter devices were purchased from Millipore (Darmstadt, Germany).

### 2.3. Experimental Design

Adult leeches of the species *H. verbana* (Annelida, Hirudinea), kindly donated by ILFARM S.r.l. (Varese, Italy), were housed in oxygenated tanks containing lightly salted water (NaCl 1.5 g/L), at a constant temperature of 20 °C. For these experiments, three animals were randomly allocated to each of the two experimental groups using a computer-generated randomization list (Microsoft Excel, Microsoft Corp., Redmond, WA, USA). Each animal was identified by a unique alphanumeric code, and group assignment was determined by referencing these codes against the randomization list to ensure unbiased and balanced distribution across groups. Each group was composed of three different animals, and for each animal, experiments were conducted in triplicate.

Group 1: 3 leeches were injected with 10 mg of MicroMatrix^®^ suspended in 30 µL of FBS-free DMEM and used as negative control (CTRL).Group 2: 3 leeches were injected with 10 mg of MicroMatrix^®^ suspended in 30 µL of hDPSC-Secretome (hDPSC-S) containing 50 µg of proteins.

Prior to injection, all animals were anesthetized by immersion in a 10% ethanol solution. All animals were inoculated using an insulin microsyringe equipped with a 22G needle. To evaluate differences in cell migration and scaffold colonization between the two experimental groups, animals were sacrificed 1-week after treatment. Prior to tissue dissection, euthanasia was performed by prolonged immersion in a 10% ethanol solution in freshwater to ensure complete and irreversible cessation of vital functions.

### 2.4. Isolation of hDPSCs and Preparation of Cell-Secretome

hDPSCs were isolated from dental pulp tissue according to the Gronthos protocol [27] modified in our laboratory [20]. hDPSC-S was prepared slightly modifying the protocol previously described [28]. Briefly, the cells were cultured until the 5th passage, once they reached 80% confluence, the medium was removed, and cells were washed twice with PBS buffer solution before adding fetal bovine serum (FBS)-free Dulbecco’s Modified Eagle Medium/Dulbecco’s Modified Eagle Medium F12 (DMEM/DMEM F12). After 72 h of starvation, the media were collected and centrifuged at 1000× *g* for 10 min, to remove cell debris. The collected secretome was concentrated using the Amicon Ultra 15 mL Centrifugal filter device with a 3 kDa cut-off, according to the manufacturer’s instructions, and total protein content was quantified using NanoOrange^®^ Protein Quantitation Kit, according to the manufacturer’s instructions (Promega, Milano, Italy). Aliquots of 50 μg of total proteins were then stored at −80 °C until use. Cell secretome characterization was performed by protein array as reported in Barone et al. 2024 [21].

### 2.5. Light and Transmission Electron Microscopy

Leech tissues were collected from the injection sites and fixed in 4% glutaraldehyde, diluted in 0.1 M cacodylate buffer (pH 7.4), overnight at 4 °C. Samples were then rinsed in the same buffer and post-fixed in 2% osmium tetroxide (OsO_4_) solution, for 1 h at room temperature. Dehydration was performed through a serial ethanol scale (50%, 70%, 90% and uranyl acetate, 100%), followed by immersion in a propylene oxide solution, for 1 h at room temperature. Samples were subsequently embedded in Epon-Araldite 812 epoxy resin.

Sections (700 nm thick) were prepared using a Reichert Ultracut S ultratome (Leica, Wein, Austria), mounted on a slide, and stained with crystal violet (1 g in 100 mL of distilled water) and basic fuchsine (0.13 g in 100 mL of distilled water). Samples were examined under a light microscope, and images acquired using a DS-5M-L1 digital camera (Nikon, Tokyo, Japan).

From the same samples, ultrathin sections (70 nm thick) were prepared and deposited on 300-mesh copper grids, counterstained with uranyl acetate and lead citrate, and examined with a JEOL1400Plus transmission electron microscope (Centro di Ricerca e Trasferimento Tecnologico [CRIETT], University of Insubria). Data were recorded using a MORADA digital camera system (Olympus, Tokyo, Japan).

### 2.6. Masson’s Trichrome and May-Grünwald Giemsa Stainings

Samples were fixed in 4% paraformaldehyde overnight at 4 °C, then washed in PBS buffer solution (138 mM NaCl, 2.7 mM KCl, 4.3 mM Na_2_HPO_4_—pH 7.4). Tissues were dehydrated with increasing ethanol concentrations (30%, 50%, 70%, 90%, 96%, 100%), cleared in pure xylol for 30 min, and embedded in paraffin.

Sections (7 µm thick) were prepared with a rotary microtome (Jung multicut 2045, Leica, Wein, Austria) and processed for either Masson’s Trichrome or May-Grünwald Giemsa staining kits, according to the manufacturer’s instructions. Samples were observed under a light microscope, and images were acquired as described above.

### 2.7. Immunofluorescence Assays

Paraffin sections, prepared as described above, were rehydrated with decreasing ethanol concentrations (100%, 96%, 70%, 30%, and water), then immersed for 30 min in citrate buffer (10 mM—pH 6) pre-heated to 95 °C for antigen retrieval, followed by three washes in PBS buffer solution. All steps were conducted at room temperature.

Samples were then pre-incubated for 30 min in BSA blocking solution (2% Bovine Serum Albumin, 0.1% Tween in PBS), which was also used for diluting both primary and secondary antibodies.

Sections were then incubated for 90 min with primary antibodies (Table 1) and, following several washes in PBS buffer solution, sections were incubated for 60 min with Cyanine3-conjugated secondary antibody (Cy3), diluted 1:200. Nuclei counterstaining was performed using 4′,6-diamidino-2-phenylinedole (DAPI, 0.1 mg/mL in PBS), and the slides mounted with Citifluor (Citifluor Ltd., London, UK). Negative controls were performed by omitting primary antibodies and incubating sections with secondary antibodies only.

All samples were analysed using a fluorescence microscope (Eclipse, Nikon, Tokyo, Japan), equipped with the emission filters 550/580 nm for Cy3 and 340/488 nm for DAPI signals. Images were acquired using a DS-5M-L1 digital camera (Nikon, Tokyo, Japan) and combined with Adobe Photoshop (Adobe System, San Jose, CA, USA).

### 2.8. Statistical Analyses

The scaffold colonization rate, the collagen fibers organization, and the fluorescence intensity area were analysed using the ImageJ software package (https://imagej.net/ij/index.html) accessed on 28 August 2025, while statistical evaluations were performed using Graph Prism 8 software (GraphPad Software, La Jolla, CA, USA; https://www.graphpad.com/features) accessed on 8 September 2025. To evaluate the precise scaffold colonization, the blue channel was isolated from RGB Masson’s Trichrome images, which were then normalized and binarized. In particular, collagen coherency was assessed using the MonogenicJ plugin (BIG-EPFL, Lausanne, Switzerland). This tool computes the local structure tensor via monogenic signal decomposition, providing quantitative measures of collagen fiber anisotropy. In contrast, the total fluorescent area was calculated by analysing ten different slides, each containing randomly selected fields of 45,000 μm^2^.

Experiments were performed in triplicate and data were reported as mean values ± standard deviation (SD). Significant differences were determined by Student’s *t*-test and a *p*-value lower than 0.05 was considered statistically significant. In the graphs, substantial differences between CTRL and hDPSC-S treated groups are indicated by asterisks as follows: * *p* < 0.05, ** *p* < 0.01, *** *p* < 0.001.

## 3. Results

### 3.1. Integration of MicroMatrix^®^ into Leech Tissues and Cellular Colonization Patterns

The ability of MicroMatrix^®^, combined with either FBS-free DMEM (CTRL) or hDPSC-S, to integrate into leech tissues was assessed using Masson’s Trichrome and May-Grünwald Giemsa stainings (Figure 1). In both conditions, the scaffolds were completely incorporated into the leech body wall. However, in CTRL samples, the scaffold showed more superficial cellular colonization (Figure 1A,C) compared to those supplemented with hDPSC-S (Figure 1B,D), which displayed cell infiltration in deeper areas as well.

May-Grünwald Giemsa staining confirmed these observations, revealing a distinct colorimetric pattern, with vasocentral cells clearly highlighted in light blue. In the CTRL condition, the innermost portion of the MicroMatrix^®^ appeared lighter (Figure 1E,G), whereas in the hDPSC-S group, a more intense blue coloration suggested higher cellular infiltration, likely associated with increased microenvironmental acidity (Figure 1F,H).

### 3.2. Evaluation of MicroMatrix^®^ Cell Colonization

Masson’s Trichrome staining, combined with quantitative image analysis, revealed significant differences in cellular colonization between the two experimental conditions. In the CTRL scaffold, cell infiltration was predominantly confined to the peripheral surface, while the inner core remained largely occupied by the blue collagen matrix (Figure 2A). By contrast, the MicroMatrix^®^ supplemented with hDPSC-S exhibited markedly greater and more uniformly distributed cell penetration throughout the scaffold, as evidenced by the larger red-stained area (Figure 2B). Quantitative measurements confirmed this pattern, showing both a significantly higher cell infiltration rate and a reduced collagen fibers organization in hDPSC-S-enriched MicroMatrix^®^ compared to CTRL ones (Figure 2C,D).

These data provide robust evidence that hDPSC-S supplementation promotes deeper and more homogeneous cellular migration within the scaffold, enhancing its integration into the host tissue.

Furthermore, although numerous cells were observed within and around the scaffold, immunofluorescence analysis for *Hm*AIF1—a marker typically expressed by activated macrophage-like cells in leeches—was performed to assess inflammatory cell presence (Appendix A, Appendix A). The results confirmed a limited number of *Hm*AIF1-positive cells, indicating that neither the control scaffold nor the hDPSC-S-enriched scaffold elicited a significant inflammatory response.

### 3.3. Phenotypic Characterization of MicroMatrix^®^-Infiltrating Cells

The effect of hDPSC-S on MicroMatrix^®^ colonization was first evaluated by light microscopy following crystal violet and basic fuchsine staining (Figure 3). MicroMatrix^®^ supplemented with hDPSC-S (Figure 3B,D) exhibited markedly higher and more homogeneous cellular infiltration compared to CTRL MicroMatrix^®^ (Figure 3A,C). Furthermore, in hDPSC-S-enriched MicroMatrix^®^, the collagen meshes appeared looser and the matrix less compact (Figure 3F) than in CTRL (Figure 3E), suggesting active ECM remodelling associated with progressive cell penetration.

Telocytes (TCs) were clearly visible, identified by their small cell bodies and long, thin cytoplasmic projections (telopodes) forming an extensive three-dimensional intercellular network. These stromal cells, located within the collagen structure, established multiple contacts with neighbouring cells, suggesting a role in coordinating cell recruitment, matrix remodelling, and spatial tissue organization (Figure 3F,G).

TEM imaging (Figure 4A,B) revealed several distinct cell populations surrounding and within MicroMatrix^®^. Elongated vasocentral cells, with electron-dense cytoplasm containing a few large granules, were mainly observed in the outer region of the scaffold. These cells were parallelly aligned and formed a thicker and more compact in hDPSC-S-enriched MicroMatrix^®^ (Figure 4B) compared to CTRL (Figure 4A).

Immunofluorescence assays (Figure 4C,D) confirmed a significantly wider CD154^+^ peripheral area in hDPSC-S-supplemented MicroMatrix^®^ (Figure 4D) compared to CTRL (Figure 4C). Quantification of CD154 fluorescence intensity (Figure 4E) showed a statistically significant increase in hDPSC-S-enriched MicroMatrix^®^, confirming enhanced recruitment of these cells relative to CTRL.

Moreover, ultrastructural analyses by TEM also revealed the presence of HPSCs within hDPSC-S-supplemented MicroMatrix^®^. These cells exhibited a blast-like morphology and an undifferentiated phenotype, characterized by a large nucleus and scant cytoplasm (Figure 5A,B).

To further characterize these cells, immunofluorescence analyses targeting CD34, a specific marker of HPSCs (Figure 5C,D), were performed; a marked increase in CD34^+^ cells was observed in hDPSC-S-supplemented MicroMatrix^®^ (Figure 5D) compared to CTRL (Figure 5C), as further confirmed by fluorescence signal quantification (Figure 5E).

Together, these data indicate that hDPSC-S promotes deeper infiltration of CD34^+^ cells within the inner compartment of the MicroMatrix^®^.

### 3.4. Pro-Angiogenic Activity and Vascular Maturation Induced by hDPSC-S

To further confirm the pro-angiogenic potential of hDPSC-S-enriched MicroMatrix^®^, TEM and immunofluorescence analyses for VEGF and CD31 were performed (Figure 6).

TEM analyses revealed newly formed blood vessels in both CTRL (Figure 6A) and hDPSC-S-enriched MicroMatrix^®^ (Figure 6B). However, vessels in the hDPSC-S-enriched MicroMatrix^®^ exhibited morphological features indicative of a more advanced maturation stage, including a continuous endothelial cell layer with tight junctions, surrounding fibroblasts, a complete and continuous basement membrane, and an organized ECM contributing to vessel wall stabilization. In contrast, vessels in the CTRL MicroMatrix^®^ displayed a thinner and, in some areas, discontinuous endothelial lining, along with an irregular or partially developed basement membrane.

The concomitant presence of both an intact endothelial barrier and a well-formed basement membrane in the hDPSC-S-enriched MicroMatrix^®^ strongly suggests improved vascular stability and integrity.

Immunofluorescence assays of VEGF and CD31 confirmed the upregulation of angiogenesis-related markers in hDPSC-S-supplemented MicroMatrix^®^ (Figure 6D,F) compared to CTRL (Figure 6C,E). In particular, VEGF staining showed a more intense and widespread distribution within and around the scaffold in the treated group (Figure 6D). Similarly, CD31 immunostaining, a well-established endothelial marker, was markedly more abundant in the hDPSC-S group (Figure 6F) than in CTRL (Figure 6E).

Quantitative analyses confirmed a significant increase in both VEGF- (Figure 6G) and CD31-positive areas (Figure 6H) in hDPSC-S-supplemented MicroMatrix^®^.

For all immunofluorescence assays performed, control samples lacking the primary antibodies showed no detectable signal, thereby demonstrating the specificity of the staining (Appendix A, Appendix A).

## 4. Discussion

Among the vascular complications associated with chronic inflammation, those that impair angiogenesis and tissue repair are particularly significant, as these processes depend on the presence of a properly structured extracellular matrix (ECM). As the mechanisms underlying the interactions between the ECM, in terms of composition, stiffness, and associated growth factors, and cellular responses are not yet fully understood, it is crucial to consider how a scaffold can influence cell behaviour. In this study, we tested the effects of MicroMatrix^®^ UBM Particulate, a nanostructured three-dimensional scaffold specifically designed to address irregular wounds and composed of collagen, laminin, and proteoglycans, on nearby cells involved in the remodelling response [18]. After 1 week from the injection of hDPSC-S-enriched MicroMatrix^®^ into the *H. verbana* body wall, we observed a significant enhancement in scaffold colonization, increased recruitment of key regenerative cell populations, and the formation of mature and stable vascular networks. The 1-week time point was selected because it represents the critical phase for detecting acute tissue responses, while longer intervals (2 weeks or 1 month) were excluded since the scaffold is typically fully integrated and no longer distinguishable from host tissue (personal observations) [29].

Histological and ultrastructural analyses revealed that hDPSC-S supplementation led to deeper and more uniform cell infiltration throughout the scaffold, markedly different from the more superficial colonization observed in the CTRL group. In addition, immunofluorescence analyses using the anti-*Hm*AIF1 antibody, a marker specific for activated macrophage-like cells in leeches, revealed a low presence of inflammatory cells, especially in scaffolds enriched with hDPSC-S. This finding indicates that the increased cellular infiltration observed is not due to an exacerbated inflammatory response, but rather to enhanced regenerative activity, consistent with previous reports [30]. This supports our working hypothesis that the bioactive factors present in hDPSC-S promote cellular recruitment and integration. Similar outcomes have been reported with other MSC-secretomes and growth factor-enriched biomatrices, further validating the synergistic effect of scaffold and secretome combinations in improving tissue regeneration [31,32]. Importantly, the recruitment of three complementary cell populations, CD154^+^ vasocentral cells, CD34^+^ Hematopoietic Precursor Stem Cells (HPSCs), and telocytes (TCs), reflects the multifaceted nature of the milieu orchestrated by hDPSC-S.

TCs play a central role in this process, extending beyond their simple structural functions. Their ability to communicate via direct contacts and paracrine signalling, including the secretion of angiogenic factors such as VEGF, positions them as key coordinators of tissue remodelling and angiogenesis [33]. This role aligns well with studies in vertebrates, where TCs regulate intercellular organization, ECM remodelling, and immune responses [34,35]. Our observation of abundant TCs forming extensive 3D networks within hDPSC-S-supplemented scaffold suggests that these cells play a pivotal role in modulating the regenerative microenvironment by coordinating the recruitment and function of both muscle precursors and endothelial cells.

The pro-angiogenic effect was evident not only through increased vessel density but also by their advanced morphological maturation observed in the hDPSC-S group. Features, such as intact endothelial layers with tight junctions, fully formed basement membranes, and organized ECM indicate structurally stable and functionally competent vasculature, which is crucial to sustain tissue repair. These findings corroborate the well-established role of VEGF as a potent endothelial chemoattractant and mitogen [36] and align with previous evidence that MSC-derived factors enhance both angiogenesis and vascular maturation in regenerative contexts [37,38]. In support of the findings reported above, we recently published the secretome content of hDPSCs [21], showing that it predominantly contains various pro-angiogenic factors such as SDF-1, IL-6, IL-8, members of the FGF protein family (FGF4, FGF6, FGF9), and VEGF-D. Interestingly, Bokhari et al. [39] reported that VEGF-D is a soluble angiogenic factor more potent than VEGF-A.

The use of *H. verbana* as an experimental model provides a simplified, yet biologically relevant, system to dissect complex regenerative interactions. Nonetheless, it is important to acknowledge species-specific limitations that may affect the direct translatability of findings to mammalian systems. In particular, the leech lacks an adaptive immune system, which is fundamental in vertebrate immune responses and tissue regeneration. This limitation restricts the model’s ability to fully replicate certain immunological mechanisms present in mammals, necessitating cautious interpretation when extrapolating regenerative and vascular outcomes. However, the conserved innate immune components and cellular crosstalk in *H. verbana* still provide valuable insights into fundamental processes of vascular maturation and tissue repair [40].

From a broader perspective, our results underscore the significance of a cell-free device composed of MicroMatrix^®^ acting as a passive architectural guide, and hDPSC-S, which provides active biological cues. This dual approach addresses the multifactorial nature of impaired wound healing, particularly in vascular-compromised tissues such as diabetic ulcers, one of the major clinical challenges that prompted this study.

Future research should aim to dissect, through proteomic and transcriptomic profiling, the specific cytokines, growth factors, and miRNAs within hDPSC-S responsible for these multifaceted effects.

Additionally, scaling these findings into mammalian models of chronic wounds or ischemia would help validate the therapeutic potential of this cell-free device and facilitate its clinical translation.

Lastly, given its conserved regenerative pathways and experimental accessibility, the *H. verbana* model could serve as a valuable screening platform for future regenerative biomaterials as cell-free therapies.

## 5. Conclusions

In conclusion, this study reinforces the concept that functionalizing an ECM-derived scaffold like MicroMatrix^®^ with hDPSC-S significantly enhances cellular infiltration, orchestrates the recruitment of diverse regenerative cell types, and promotes the development of mature vascular networks, ultimately accelerating the wound healing process. These results validate the therapeutic potential of the cell-free device composed of MicroMatrix^®^ and hDPSC-S, positioning it as a promising approach for regenerative treatments targeting complex, poorly vascularized wounds. Furthermore, this model provides a solid foundation for future mechanistic studies exploring tissue regeneration and angiogenesis.

## Figures and Tables

**Figure 1 micromachines-16-01150-f001:**
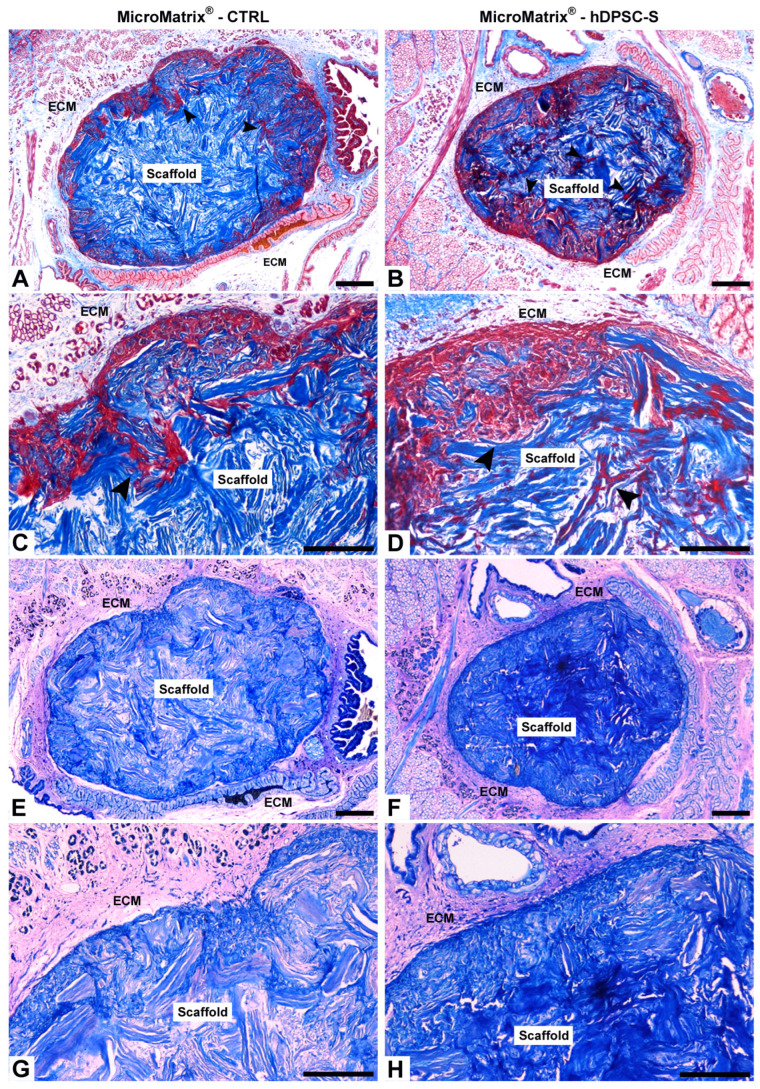
Integration of MicroMatrix^®^ scaffold into leech tissues. Masson’s Trichrome (**A**–**D**) and May-Grünwald Giemsa (**E**–**H**) staining show the integration of both CTRL and hDPSC-S scaffolds into the leech body wall. In CTRL (**A**,**C**), cell colonization (black arrowheads) is superficial, whereas in MicroMatrix^®^-hDPSC-S (**B**,**D**), deeper infiltration is observed. Giemsa staining confirms these findings for both CTRL (**E**,**G**) and MicroMatrix^®^-hDPSC-S (**F**,**H**), with the latter displaying a more intense blue, indicating higher cell infiltration. ECM: extracellular matrix. Bars in (**A**,**B**,**E**,**F**): 100 µm; bars in (**C**,**D**,**G**,**H**): 50 µm.

**Figure 2 micromachines-16-01150-f002:**
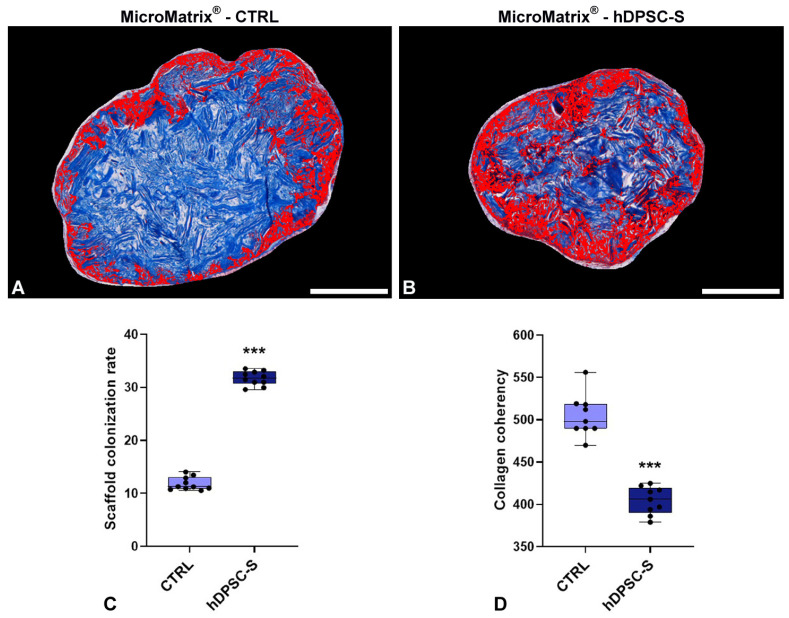
MicroMatrix^®^ cellular colonization rate and collagen fibers spatial organization. Image-based analyses conduced on Masson’s Trichrome staining (**A**,**B**) reveal deeper and more homogeneous infiltration in MicroMatrix^®^-hDPSC-S, as indicated by the expanded red-stained region and also confirmed by the quantitative analyses presented in the graph (**C**). In addition, a reduced collagen fibers coherency is detectable in hDPSC-S-supplemented scaffold (**D**), suggesting that infiltrated cells actively remodel the inner matrix. Statistical significance between CTRL and MicroMatrix^®^-hDPSC-S samples was calculated using Student’s *t*-test, and bars represent mean ± standard deviation (SD). *** *p* < 0.001. Bars in (**A**,**B**): 100 µm.

**Figure 3 micromachines-16-01150-f003:**
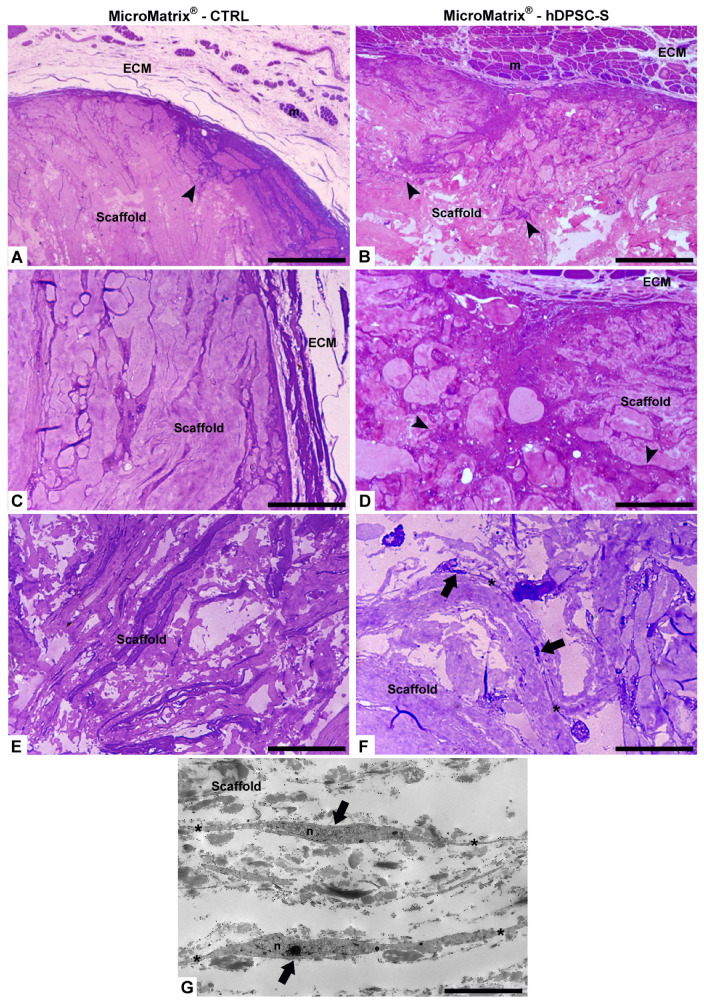
Morphological analyses by optical and TEM microscopy. Compared to CTRL (**A**,**C**,**E**), a more homogeneous cellular infiltration is visible in MicroMatrix^®^-hDPSC-S (**B**,**D**,**F**), where collagen fibers appear looser and less compact, indicating active ECM remodelling and cellular penetration (black arrowheads). Telocytes (black arrows in (**F**,**G**)) are characterized by small cell bodies and long telopodes (*). ECM: extracellular matrix; m: muscles fibers; n: nucleus. Bars in (**A**–**F**): 100 µm; bar in (**G**): 5 µm.

**Figure 4 micromachines-16-01150-f004:**
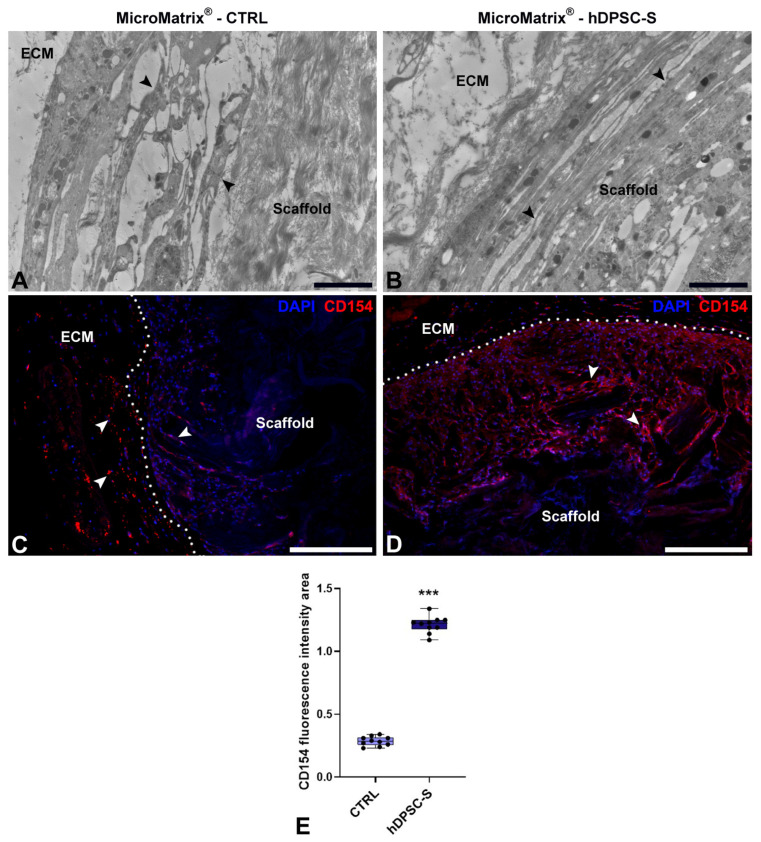
Characterization of cell populations in MicroMatrix^®^. TEM images (**A**,**B**) reveal elongated vasocentral cells (black arrowheads) with electron-dense cytoplasm located outside the scaffold. These cells are more compact and aligned in the MicroMatrix^®^-hDPSC-S sample (**B**) compared to CTRL (A). Immunofluorescence staining (**C**,**D**) indicates a higher CD154^+^ signal (white arrowheads) in MicroMatrix^®^-hDPSC-S (**D**), as demonstrated by the fluorescence intensity graph (**E**). Dashed lines in (**C**,**D**) delineate the boundaries between scaffold and ECM. Statistical significance between CTRL and MicroMatrix^®^-hDPSC-S was calculated using Student’s *t*-test, and bars represent mean ± standard deviation (SD). *** *p* < 0.001. ECM: extracellular matrix. Bars in (**A**,**B**): 10 µm; bars in (**C**,**D**): 50 µm.

**Figure 5 micromachines-16-01150-f005:**
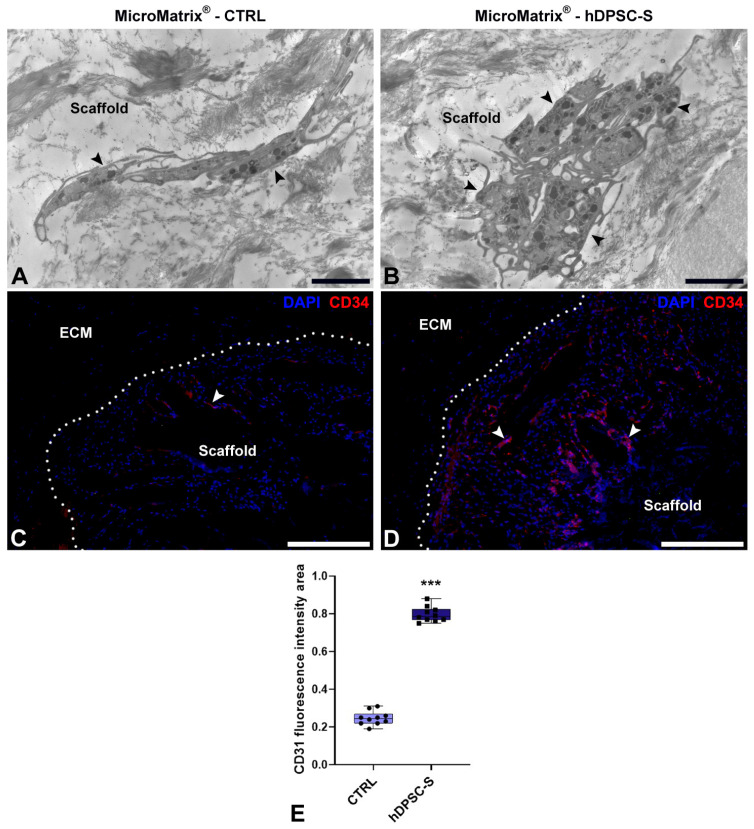
Hematopoietic Stem Precursor Cells (HSPCs) recruitment in MicroMatrix^®^. Compared to CTRL (**A**), TEM analyses reveal numerous HSPCs (black arrowheads) within the MicroMatrix^®^-hDPSC-S (**B**). Immunofluorescence assays (**C**,**D**) confirm this result, showing a marked increase in CD34^+^ cells (white arrowheads). Dashed lines in (**C**,**D**) delineate the boundaries between scaffold and ECM. Quantification of the CD34 signal (**E**) validates this result. Statistical significance between CTRL and MicroMatrix^®^-hDPSC-S samples was calculated using Student’s *t*-test, and bars represent mean ± standard deviation (SD). *** *p* < 0.001. ECM: extracellular matrix. Bars in (**A**,**B**): 10 µm; bars in (**C**,**D**): 50 µm.

**Figure 6 micromachines-16-01150-f006:**
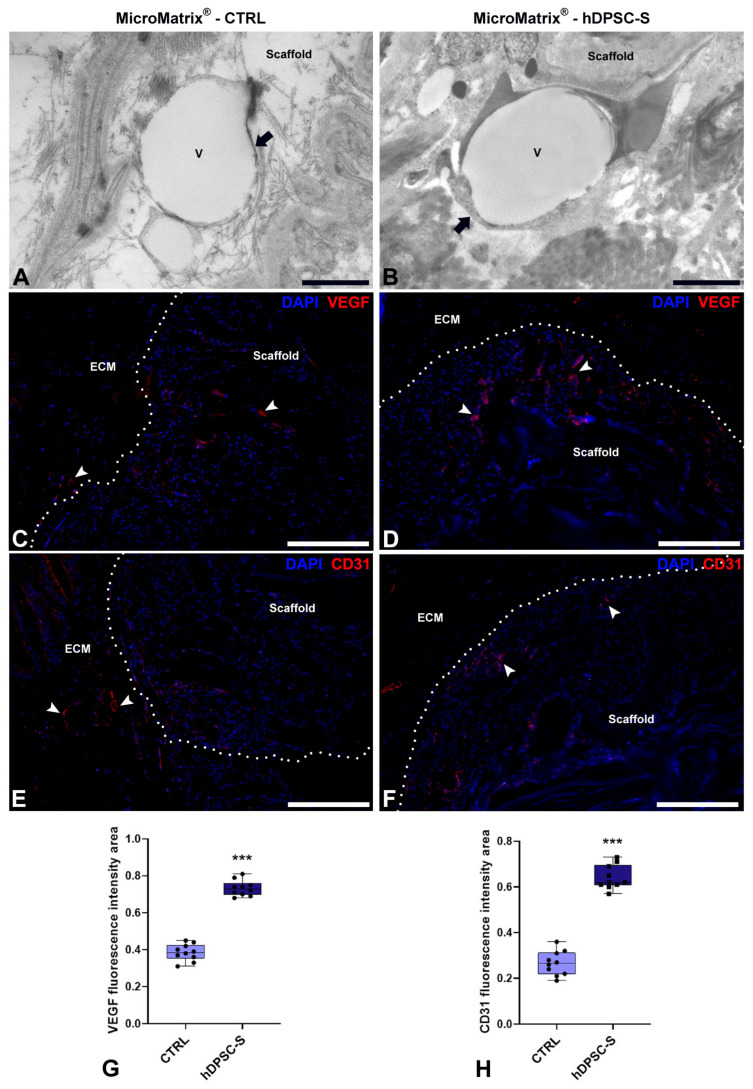
Pro-angiogenic effects of hDPSC-S in MicroMatrix^®^. TEM analyses (**A**,**B**) reveal the presence of newly formed blood vessels (black arrows) in both CTRL (**A**) and MicroMatrix^®^-hDPSC-S (**B**). Vessels in the MicroMatrix^®^-hDPSC-S display a more advanced maturation stage, with a continuous endothelium and well-organized ECM. Immunofluorescence staining (**C**–**F**) shows higher signals for both VEGF and CD31 (white arrowheads) in MicroMatrix^®^-hDPSC-S (**D**,**F**) than in CTRL (**C**,**E**), as demonstrated by quantitative analyses (**G**,**H**). Dashed lines in C–F delineate the boundaries between scaffold and ECM. Statistical significance between CTRL and MicroMatrix^®^-hDPSC-S was calculated using Student’s *t*-test, and bars represent mean ± standard deviation (SD). *** *p* < 0.001. ECM: extracellular matrix. Bars in (**A**,**B**): 10 µm; bars in (**C**–**F**): 50 µm.

**Table 1 micromachines-16-01150-t001:** Primary antibodies used in this work.

Antibody	Description	Company	Application	Dilution
CD31	α-human mouse monoclonal	Novocastra (Nussloch, Germany)	IF	1:200
CD34	α-human rabbit polyclonal	Abcam (Cambridge, UK)	IF	1:100
CD154	α-human rabbit polyclonal	Santa Cruz Biotechnology (Texas, TX, USA)	IF	1:100
VEGF	α-human mouse monoclonal	ThermoFisher Scientific (Waltham, CA, USA)	IF	1:100
*Hm*AIF1	α-*Hirudo medicinalis* rabbit polyclonal	Kindly donated by Prof. Vizioli (University of Lille, France)	IF	1:100

## Data Availability

Raw data are available, upon request, from the corresponding authors.

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
