# Peer review of "Nanostructured Scaffold, Combined with Human Dental Pulp Stem Cell Secretome, Induces Vascularization in Medicinal Leech Model"

_micromachines, 2025, doi:10.3390/mi16101150_

Round 1
Reviewer 1 Report
Comments and Suggestions for Authors
This study investigates the regenerative potential of MicroMatrix® UBM Particulate, a nanostructured extracellular matrix (ECM) scaffold derived from porcine urinary bladder, either alone (CTRL) or supplemented with the secretome from human dental pulp stem cells (hDPSC-S), using the medicinal leech Hirudo verbana as an in vivo model for wound healing and vascularization. Adult leeches were injected with the scaffold formulations, sacrificed after one week, and analyzed via histological stainings (Masson’s Trichrome, May-Grünwald Giemsa), transmission electron microscopy (TEM), and immunofluorescence for markers like CD154 (vasocentral cells), CD34 (hematopoietic precursor stem cells, HPSCs), VEGF, and CD31 (endothelial cells). Results demonstrated that hDPSC-S supplementation significantly enhanced scaffold integration, promoted deeper and more homogeneous cellular infiltration including telocytes, CD154+ vasocentral cells, and CD34+ HPSCs, and induced higher vessel density with more mature vascular structures characterized by intact endothelial layers, basement membranes, and organized ECM, highlighting the cell-free device's efficacy in accelerating angiogenesis and tissue repair for potential applications in treating poorly vascularized wounds like diabetic ulcers.
- In the methods section, you mention using three leeches per group with experiments conducted in triplicate. Could you clarify how the triplicates were performed—were they biological replicates (independent leech injections) or technical replicates (multiple analyses from the same samples)? This would help assess the robustness of the statistical analyses.
- The hDPSC-S was concentrated to 50 μg of total proteins per aliquot, but no details are provided on the specific composition. Could you provide more information on the secretome's characterization, as this would strengthen the mechanistic insights into its pro-angiogenic effects?
- In the results, TEM shows more mature vessels in the hDPSC-S group, but functional assessments (e.g., perfusion or blood flow) are absent. Could you clarify if any in vivo functional assays were attempted, or why they were not included in this model?
- The discussion mentions parallels with vertebrate mechanisms but does not address potential species-specific differences in the leech model (e.g., lack of adaptive immunity). Could you elaborate on how these might limit translatability to mammalian systems?
- The introduction effectively discusses the role of ECM substitutes but could be strengthened by comparing MicroMatrix® to other nanostructured scaffolds used in tissue regeneration. For instance, include references to innovative materials 10.1021/acsami.6b14836. This would provide a broader context and highlight unique advantages of your porcine-derived scaffold.
- While ImageJ was used for quantification, consider including more detailed metrics, such as vessel diameter, branching index, or infiltration depth distributions (e.g., via histograms). Additionally, use box plots instead of bar graphs for fluorescence intensity data to better represent variability and outliers.
- The one-week endpoint provides valuable early insights, but suggest extending to longer time points (e.g., 2-4 weeks) to assess sustained vascular stability, potential scaffold degradation, or full tissue remodeling, which would better mimic chronic wound healing.
- To deepen the discussion, include preliminary data or future plans for RNA-seq or proteomics on infiltrated cells to identify hDPSC-S-induced pathways (e.g., HIF-1α or Notch signaling), linking morphological changes to molecular events.
- Figures are informative, but add scale bars to all panels consistently and include negative controls for immunofluorescence (e.g., isotype-matched antibodies) in the figures. Also, annotate key cellular features (e.g., telopodes) more prominently in TEM images for better accessibility.
Author Response
Please see the attacement

Reviewer 2 Report
Comments and Suggestions for Authors
Authors added hDPSC Secretome (hDPSC-S) in MicroMatrix® to enhance ECM remodelling. This is a solid work, from material design to result presentation. I generally agree with the publication of this work. I have a few questions:
1. How many generations can the hDPSC maintain their stemness properties?
2. In “Group 2: 3 leeches were injected with 10 μg of MicroMatrix® suspended in 30 μL of 132 hDPSC-Secretome (hDPSC-S) containing 50 μg of proteins.” The mass of proteins is much more than that of MicroMatrix®. Have you ever tried to inject the hDPSC-S alone into the leeches?
Reviewer 3 Report
Comments and Suggestions for Authors< !--StartFragment -->
In this article, the authors have evaluated a cell-free wound healing device composed of MicroMatrix® and the secretome of hDPSCs for its potential to treat wounds with poor vascularization. While the study presents novel observations, the authors are requested to address the following issues to improve the manuscript.
- The authors are requested to clarify the wound model by providing details, including how the wounds were created and the rationale behind selecting the concentration of MicroMatrix and hDPSC-secretome.
- The authors only selected one time point (day 7) as the endpoint, which is not sufficient to capture the differences in the wound healing process between the hDPSC and MicroMatrix groups. Moreover, the experimental design lacks a "sham group," a critical control needed to evaluate and compare the outcomes of both treatment groups. The authors are requested to address these critical issues.
- As reported in Figure 1 (E, F) and Figure 2, the more intense blue coloration in the hDPSC-S group might indicate increased cellular infiltration, potentially due to a more severe inflammatory response compared to the control group. The authors are requested to provide further clarification on this phenomenon.
- The authors mentioned that "in hDPSC-S-enriched MicroMatrix®, the collagen meshes appeared looser and the matrix less compact (Figure 3F) than in CTRL (Figure 3E), suggesting active ECM remodeling associated with progressive cell penetration." To support this claim, cell nuclei must be visible in the images, which is currently not the case. The authors could perform H&E staining to clearly demonstrate an increased number of cell nuclei in the treatment group. Furthermore, to confirm that the marked cells are indeed telocytes, IF-staining with antibodies targeting telocyte-specific cell surface markers is required.
- Increased CD154 signal in the wound area might indicate enhanced infiltration of immune cells in the wound matrix. Although the authors have provided CD34+ staining data to suggest that a portion of the infiltrated cells could be HPSCs, clear immune cell–specific staining is necessary to ensure that hDPSC-S treatment does not induce a severe pro-inflammatory response.
- The CD31 signal between the control and experimental groups does not show a drastic difference in Fig. 6 (E, F). The authors are requested to clarify how the quantification of CD31 signal was calculated (i.e., number of animals per treatment group, number of stained sections per animal, number of areas analyzed within the sections). The bar graph shows only six data points, which appears insufficient to provide adequate statistical power, especially if only three animals per treatment group were included. Moreover, the Y-axis on all bar graphs does not provide units to define the "area."
- Overall, the authors relied only on image-based analysis and did not perform any biochemical or molecular biology–based assays (e.g., RT-qPCR for inflammation, angiogenesis, and proliferation-associated markers) to provide clear evidence that the hDPSC-S group had superior healing effects compared to the CTRL group.
< !--EndFragment -->
Round 2
Reviewer 1 Report
Comments and Suggestions for Authors
The authors have effectively addressed all the comments previously raised and the revised manuscript is now suggested for publication.
Reviewer 3 Report
Comments and Suggestions for Authors
Authors have satisfactorily addressed the comments.